



# Simulated Multispectral Temperature and Atmospheric
# Composition Retrievals for the JPL GEO-IR Sounder
Vijay Natraj[1], Ming Luo[1], Jean-Francois Blavier[1], Vivienne H. Payne[1], Derek J.
Posselt[1], Stanley P. Sander[1], Zhao-Cheng Zeng[2,3], Jessica L. Neu[1], Denis Tremblay[4],
Longtao Wu[1], Jacola A. Roman[1], Yen-Hung Wu[1], Leonard I. Dorsky[1]
[1]Jet Propulsion Laboratory, California Institute of Technology, Pasadena, CA 91109, USA
[2]Joint Institute for Regional Earth System Science and Engineering, University of California, Los
Angeles, CA 90095, USA
[3]Division of Geological and Planetary Sciences, California Institute of Technology, Pasadena, CA 91125,
USA
[4]Global Science Technology, USA
*Correspondence to*: Vijay Natraj (vijay.natraj@jpl.nasa.gov)
**Abstract.** Satellite measurements enable quantification of atmospheric temperature, humidity, and trace
gas vertical profiles. The majority of current instruments operate on polar orbiting satellites and either in
the thermal/mid-wave or in the shortwave infrared spectral regions. We present a new multispectral
instrument concept for improved measurements from geostationary orbit (GEO) with sensitivity to the
boundary layer. The JPL GEO-IR sounder, which is an imaging Fourier Transform Spectrometer, uses a
wide spectral range (1–15.4 μm), encompassing both reflected solar and thermal emission bands to
improve sensitivity to the lower troposphere and boundary layer. We perform retrieval simulations for
both clean and polluted scenarios that also encompass different temperature and humidity profiles. The
results illustrate the benefits of combining shortwave and thermal infrared measurements. In particular,
the former adds information in the boundary layer, while the latter helps to separate near-surface and
mid-tropospheric variability. The performance of the JPL GEO-IR sounder is similar to or better than
currently operational instruments. The proposed concept is expected to improve weather forecasting,
severe storm tracking and forecasting, and also benefit local and global air quality and climate research.


## 1 Introduction

The Program of Record (PoR) of current and planned satellite observations, as described in the 2017 US Earth Science Decadal Survey (*NASEM*, 2018), includes a range of spectrally-resolved radiance measurements in the thermal and shortwave infrared (TIR and SWIR) wavelength regions that provide key information on atmospheric temperature (TATM), water vapor ($H_2O$) and a range of trace gases (see Table 1 for a definition of spectral range designations). The TIR region can be further subdivided into midwave, longwave and very longwave infrared (MIR, LWIR and VLWIR) regions. Profiling of key gases including CO, $CH_4$, and $CO_2$ with sensitivity to planetary boundary layer (PBL) abundances was identified as a gap in current capability in the 2017 Decadal Survey, as was the promise of multispectral approaches for addressing this gap. In fact, combining radiances from the (thermal emission dominated) TIR and (solar reflection dominated) SWIR spectral regions has been shown to increase the vertical information content for these gases, providing improved information on near-surface variations relative to retrievals from the thermal alone (e.g., *Christi and Stephens*, 2004; *Worden et al.*, 2010; *Kuai et al.*, 2013; *Worden et al.*, 2015; *Fu et al.*, 2016; *Zhang et al.*, 2018; *Schneider et al.*, 2021). Such retrievals have the potential to extend the utility of satellite products for air quality forecasting, greenhouse gas monitoring and carbon cycle research. In addition, combining TIR and SWIR infrared radiances also offers opportunities for increasing the vertical information of $H_2O$ retrievals in the PBL, another topic highlighted by the Decadal Survey and by the NASA Decadal Survey PBL Incubation Study Team (*Teixeira et al., 2021*). Under clear-sky conditions, the SWIR provides sensitivity to $H_2O$ (e.g., *Noël et al.*, 2005; *Trent et al.*, 2018; *Nelson et al.*, 2016), CO (e.g., *Buchwitz et al.*, 2004; *Deeter et al.*, 2009; *Landgraf et al.*, 2016; *Borsdorff et al.*, 2017; 2018), $CH_4$ (e.g., *Buchwitz et al.*, 2005; *Frankenberg et al*, 2006; *Yokota et al.*, 2009; *Hu et al.*, 2018; *Parker et al*., 2020) and $CO_2$ (e.g., *Buchwitz et al.*, 2005; *Yokota et al*., 2009; *O'Dell et al*., 2018) throughout the full atmospheric column, providing complementary information to the TIR radiances that are strongly sensitive to the details of the profile of TATM, $H_2O$ and trace gases but have variable sensitivity to the PBL, depending on surface and atmospheric conditions.

Table 2 shows a list of current and planned missions making spectrally-resolved, spaceborne TIR and SWIR measurements. In Low Earth Orbit (LEO), the MOPITT instrument on the Terra platform has been providing a record of TIR+SWIR CO for over two decades (*Buchholz et al.*, 2021). GOSAT and GOSAT-2 provide spectrally-resolved TIR and SWIR radiances on the same platform, with coverage of SWIR $CO_2$ and $CH_4$ bands, as well as $H_2O$ absorption (*Trent et al.*, 2018), but not SWIR CO. The TROPOMI instrument on the Sentinel 5P satellite flies in formation with the CrIS instrument on the S-NPP satellite, providing near-coincident observations of TIR and SWIR, presenting opportunities for

multispectral retrievals of CO and CH$_4$. Measurements from geostationary (GEO) orbit can provide
contiguous horizontal (~4 km) and temporal (1–4 hours) resolution not possible from LEO. None of the
current instruments/missions listed in Table 2 provide TIR + SWIR measurements from GEO on the
same platform.

Here, we describe an instrument concept, called the JPL GEO-IR Sounder, that would provide

profiling of TATM, H$_2$O, CO, CH$_4$ and CO$_2$, as well as numerous other species important for air quality
and the hydrological cycle, from a geostationary platform. The JPL GEO-IR Sounder is an imaging
Fourier transform spectrometer that utilizes high-speed digital focal plane arrays to record simultaneous
TIR and SWIR spectra from each pixel of the array (640 ×480 or 1024×1024 format). The primary
advantages of this sounder include the following:
• Coincident spatial and temporal retrievals of trace gases and TATM using both SWIR and TIR bands

multiple times per day

• Combined TIR and SWIR retrievals provide for enhanced vertical resolution with PBL visibility for

TATM, humidity and multiple trace gases

• Capability for retrievals of 4-D winds from combinations of cloud and H$_2$O temporal imagery as

recently described using GIIRS data (*Ma et al*., 2021)

• Providing data products that are not readily obtained by combining retrievals from PoR LEO and

GEO sounders.

This paper is organized as follows: in Section 2, we describe the scenarios used in the simulations.

Section 3 provides brief descriptions of the radiative transfer (RT), instrument and inverse models. We
discuss the considerations imposed on simulated JPL GEO-IR Sounder retrievals in Section 4. In Section
5, we present results for TATM, H$_2$O and trace gas retrievals from simulated GEO-IR Sounder
measurements for both individual spectral regions and combinations. The relevance of these simulated
retrievals for Observation System Simulation Experiments is discussed in Section 6. We arrive at some
preliminary conclusions in Section 7. In particular, we show that the JPL GEO-IR Sounder would, for
the first time, enable high spatial and temporal resolution simultaneous retrievals in the TIR and SWIR,
which together provide more vertical profile information and improved sensitivity to the PBL than either
spectral region alone.

**2 Scenarios**

Representative atmospheric conditions, including TATM, H$_2$O and pollutant distributions, surface

temperature and other interferents are needed to understand satellite instrument performance. Using
Weather Research and Forecasting model coupled to Chemistry (WRF-Chem) simulations at 4 km spatial



resolution over the continental United States (Mary Barth, personal communication), we examined about
200 atmospheric profiles at six local times for two days in July 2006 over 17 locations that represent a
range of diurnal meteorological conditions and a variety of air quality scenarios. For the purposes of
these simulations, we assume clear-sky conditions. Simulation of conditions with significant aerosol
loading and cloud interference adds significant complexity and is beyond the scope of this study. We
calculate molecular absorption coefficients using the Line-By-Line Radiative Transfer Model
(LBLRTM; *Clough et al.*, 2005).
The main goal of these simulations is to evaluate the retrieval characteristics of TATM, $H_2O$, and
trace gases for different instrument configurations. From our database of over 200 summer-time
atmospheric profiles over the continental US, we selected two representative daytime atmospheres; one
near Houston to support the weather-focused OSSE analyses and the background trace gas case, and
another in West Virginia that has more enhanced trace gas pollutants near the surface. Note that we kept
the solar and viewing geometry as well as the surface albedo constant in order to isolate the effects of
different boundary layer trace gas concentrations. Figure 1 shows the profile plots for TATM, $H_2O$, and
trace gases that we examine in this manuscript ($O_3$, CO, $CH_4$ and $CO_2$) at the two locations.
The emissivity is obtained from a database structured by month and latitude/longitude coordinates. To
populate the database, we used a global land use and land cover classification system developed by the
U.S. Geological Survey (*Anderson et al.*, 1976) and mapped them into spectra from the ECOSTRESS
spectral library (*Baldridge et al.*, 2009; *Meerdink et al.*, 2019; http://speclib.jpl.nasa.gov/), as described
in the TES Algorithm Theoretical Basis Document (*Beer et al.*, 2002). The albedo is calculated from the
emissivity using Kirchoff's law.
The location and times of the WRF-Chem profiles were used to calculate the solar viewing geometry,
assuming a geostationary satellite at 95 W. The NOAA solar position calculator was used to verify the
solar zenith and solar azimuth calculations (http://www.srrb.noaa.gov/highlights/sunrise/azel.html).

**3 Models**
**3.1 Radiative transfer model**
We use the accurate and numerically efficient two-stream-exact-single-scattering (2S-ESS) RT
model (*Spurr and Natraj*, 2011; *Xi et al.*, 2015). This forward model is different from a typical two-
stream model in that the two-stream approximation is used only to calculate the contribution of multiple
scattering to the radiation field. Single scattering is treated in a numerically exact manner using all
moments of the scattering phase function. High computational efficiency is achieved by employing the
two-stream approximation for multiple scattering calculations. The exact single scattering calculation



largely eliminates biases due to the severe truncation of the phase function inherent in a traditional two-
stream approximation. Therefore, the 2S-ESS model is much more accurate than a typical two-stream
model, and produces radiances and Jacobians that are typically within a few percent of numerically exact
calculations and in most cases with biases much less than a percent. This model has been widely used
for the remote sensing of greenhouse gases and aerosols (*Xi et al.*, 2015; *Zhang et al.*, 2015, 2016; *Zeng*
*et al.*, 2017, 2018). Aerosols are not included in the analysis since the main objective was to investigate
the impact of combining multiple spectral bands and of varying instrument parameters. However, the RT
model has the capability of handling generic aerosol types.
The 2S-ESS RT model is used to generate monochromatic radiances at the top of the atmosphere
for the atmospheric profiles and surface conditions near Houston over the entire spectral range considered
for the JPL GEO-IR Sounder. Figure 2 shows the spectral radiance computed on a 0.002 cm$^{-1}$ wavelength
grid. We also calculate the individual contributions of each absorbing gas to the radiance. The gaseous
absorption features have different spectral distributions and line strengths, which can be used to identify
spectral windows for profile retrievals and recognize interfering gases that also absorb strongly in the
same channels.
**3.2 Instrument model**
This section starts with a brief description of the spectrometer, primarily to define the terms used in
the instrument model. We then detail the focal plane arrays and the optical filter that determine the
bandpasses of the instrument. The processing steps of the instrument model are then explained. Finally,
we show some of the resulting spectra produced by the model.
*3.2.1 Optics Overview*
The JPL GEO IR Sounder uses a Michelson interferometer, which modulates the light that passes
through it. The interferometer is characterized by two main parameters: the spectral resolution, which is
directly proportional to the maximum optical path difference (MOPD) between the two arms of the
interferometer, and the optical throughput or étendue, which is given by the product of the area of the
aperture stop and the angular field of view (AΩ). From geostationary orbit, a ground pixel of 2.1 km
subtends an angle of 58.7 µrad and for a Focal Plane Array (FPA) of 1024×1024 pixels, the overall FOV
is 60 mrad; this fits well within the Fourier Transform Spectrometer (FTS) design parameters. In parallel
with the light from the target scene, a beam from an internal metrology laser travels through the
interferometer. This laser is used to precisely measure the optical path difference, to within a small
fraction of the laser wavelength. An imaging FTS (IFTS) shares many of the principles of the traditional
FTS, the main difference is that the detector is replaced with an FPA. The main challenge in the IFTS



design is in the FPA, which must operate at high frame rate (0.5–1 kHz) and at high dynamic range (14–
16 bits) to properly digitize the interferograms.
*3.2.2 Focal Plane Arrays*
The JPL GEO-IR Sounder FPA optics uses a dichroic to split the interferometer output along the
wavelength dimension: radiation from 1 μm to 5.3 μm is sent towards FPA #1 and radiation from 5.3 μm
to 15.4 μm is directed to FPA #2. Whereas FPA #2 is a single-color detector, handling its full domain at
all times, FPA #1 is a dual-color detector. The two colors of FPA #1 are operated sequentially: recording
either the 1 μm to 3 μm domain (SWIR; FPA #1a) or the 3 μm to 5.3 μm domain (MWIR; FPA #1b).
This dual-color operation is implemented inside the FPA by having two distinct detectors in an optical
"sandwich". It is designed to minimize the effect of photon noise in the low-light MWIR and SWIR
domains. Furthermore, the SWIR FPA #1a bandpass is narrowed by a triple-band optical filter, tailored
to the regions that contain absorption bands of interest (Figure 3). As listed in Table 3, the SWIR domains
of interest are: (1) 4210–4350 $cm^{-1}$, (2) 4810–4900 $cm^{-1}$, (3) 6000–6150 $cm^{-1}$ and (4) 6170–6290 $cm^{-1}$.
Based on previous optical filter studies, we allow 200 $cm^{-1}$ for the filter slope on either side. Since the
gap between the first two domains would therefore be small, and the signal there is low, these have been
merged (4210–4900 $cm^{-1}$). Domains (3) and (4) have also been combined (6000–6290 $cm^{-1}$). In addition,
the 1.27 μm oxygen band (7780–8010 $cm^{-1}$) will be used to measure the light path. We believe that it is
best to specify the 50% transmission points for the filter bands, as that is where the slope is maximum
and hence most easily verified. With a 200 $cm^{-1}$ transition region, the 50% point will be 100 $cm^{-1}$ outside
of the bandpasses. Hence the final triple-band filter configuration is: 4110–5000 $cm^{-1}$ (2.000–2.433 μm),
5900–6390 $cm^{-1}$ (1.565–1.695 μm), 7680–8110 $cm^{-1}$ (1.233–1.302 μm). The triple-band filter physically
covers the two-color FPA #1. It is intended to limit the photon flux only in the SWIR mode of operation,
with the detector that is sensitive over the 1–3 μm domain (FPA #1a). The filter must also be transparent
over the 3–5.3 μm domain of the other shared detector (FPA #1b). It may be possible to combine the
first band of the triple-band filter (2–2.433 μm) with this MWIR transparency need (3–5.3 μm) but this
has not been simulated in this study.
*3.2.3 Instrument Model Description*
The Instrument Model for the JPL GEO IR Sounder allows us to explore the instrument trade space
and its effect on retrieved atmospheric composition. It includes the ability to convolve synthetic spectra
and Jacobians with the instrument line shape (ILS). The model performs the following steps:
1. Reads synthetic data from the radiative transfer model. The radiance spectrum is extended using
blackbody curves simulating the Earth and the Sun, and converted to a photon flux spectrum. After this
step, the spectrum is in units of photons/$m^2$/sr/$cm^{-1}$/s.





2. Convolves the spectrum with the theoretical FTS ILS, given as: $2L\text{sinc}(2\sigma L)$, where L is the MOPD
and $\text{sinc}(x)=\sin(\pi x)/\pi x$. This expression of the ILS has unit area, and hence the convolution does not
change the overall magnitude or the units of the spectrum. It does, however, reduce the spectral resolution,
broadening all sharp features. In the same step, we resample the spectrum on a coarser grid, i.e., we
"decimate" the spectrum. For example, in the current simulations, we reduce the wavenumber interval
by a factor of 50, from 0.002 cm$^{-1}$ to 0.1 cm$^{-1}$.
3. Scales the spectrum by the étendue of the instrument. After this step, the units of the spectrum are
photons/cm$^{-1}$/s.
4. Applies further scaling to account for the single output design (where half the light is sent through the
instrument and the other half sent back to the source), losses in the metallic coatings and at the uncoated
optical interfaces (i.e., compensator and back side of beamsplitter), the efficiency of the beamsplitter
coating, the quantum efficiency of the detector, and the integration time of the analog-to-digital converter.
After this step, the units of the spectrum are photoelectrons/cm$^{-1}$.
5. Applies bandpass limits caused either by an optical filter or the working domain of the detector.
6. Applies the Fourier transform to convert the spectrum into an interferogram.
7. Computes the number of photoelectrons counted in each interferogram data sample. From this, we can
compute the photon noise. Subsequently, white noise is added to the interferogram with a root mean
square amplitude matching the computed photon noise.
8. Simulates the interferogram digitization, performed for each pixel within the Read Out Integrated
Circuit of the two FPAs.
9. Produces the final spectrum by Fourier transform. The signal to noise ratio (SNR) is then evaluated
by computing the noise level in blacked-out regions on either side of the instrument bandpass, and by
locating the maximum signal within the bandpass.
*3.2.4 Spectral Results*

Figure 3 shows a JPL GEO-IR Sounder model spectrum for FPA #1a, covering the SWIR domain.

Figure 4 shows a similar spectrum for the VLWIR, LWIR and MWIR FPA bands: FPA #2 covers the
VLWIR and LWIR domains, and FPA #1b covers the MWIR domain. The spectral ranges include the
range utilized by existing TIR sounders (AIRS, CrIS, IASI) and selected bands in the SWIR. In particular,
the FPA #2 spectral range contains critical information for radiance assimilation by weather forecasting
algorithms (see, e.g., *Eresmaa et al.*, 2017). The spectral resolution (MOPD) of the JPL GEO-IR Sounder
is configurable. For these simulations, we choose to look at three possible MOPD options: a CrIS-like
spectral resolution (0.8 cm MOPD, 0.625 cm$^{-1}$ resolution, described as nominal spectral resolution or
NSR in Table 4), an intermediate option (2 cm MOPD, 0.25 cm$^{-1}$ resolution), and a high spectral
resolution option (5 cm MOPD, 0.1 cm$^{-1}$ resolution, described as full spectral resolution or FSR in Table
4). In order to make for an "apples to apples" comparison, we consider the same integration time (1
millisecond per interferogram point) for these three options. The integration time is driven by the high
spectral resolution option. The native and binned (footprint-averaged) ground sampling distance (GSD)
are also indicated in Table 4.
**3.3 Inverse model**
We use an optimal estimation approach (*Rodgers*, 2000) and perform linear retrievals from
simulated radiances described in the previous section. The spectral differences of the modeled and the
satellite measured radiances and the differences of the species profile and the *a priori* profile are
mathematically minimized, weighted by the measurement error and the *a priori* constraint. The species
profile can then be derived optimally.
The *a priori* constraint vectors for TATM and $H_2O$ are obtained from forecast fields from the NASA
Global Modeling and Assimilation Office, supplied for use within the TES retrieval algorithm (*Bowman*
*et al.*, 2006). *A priori* constraint matrices are constructed, using the method described in *Kulawik et al.*
(2006), from an altitude-dependent combination of zeroth, first and second order derivatives of the
profiles. For TATM and $H_2O$, the square roots of the diagonals of the respective constraint matrices are
on the order of 1.8–2.2 K and 15–18%, respectively. *A priori* vectors for $O_3$, CO and $CH_4$ are taken from
calculations using the Model for OZone And Related chemical Tracers (MOZART3) (*Brasseur et al.*,
1998; *Park et al.*, 2004) that were performed for the purpose of construction of trace gas climatologies
for the Aura mission. For $O_3$, the square root of the diagonal of the constraint matrix is on the order of
25% in the troposphere, 40% in the stratosphere and 15% above. For CO, this is set to 30% over the
entire atmosphere, while for $CH_4$, the values range from 2–10%. The constraint matrices for CO are the
same as those used by the MOPITT algorithm (*Deeter et al.*, 2010). For $CO_2$, the *a priori* vector and
constraint used are described in *Kulawik et al.* (2010). The square root of the diagonal of the constraint
matrix ranges from 1.2–2%. We note that these profile constraints were developed for TIR instruments,
and may therefore not capture strong near-surface variability. There could be scope for increasing the
near-surface information content via development of updated constraints, although that work is outside
of the scope of this study.
The end-to-end retrieval analysis provides averaging kernels, which describe the sensitivity of the
retrieved atmospheric state to the true state; degrees of freedom for signal (DOFS), which denote the
pieces of vertical information contained in the retrieved profile; and retrieval errors. These metrics are
used for evaluating the retrieval results for a variety of spectral bands, and spectral and spatial resolutions.

## 4 Considerations for simulated retrievals

For the retrieval simulations described here, we consider a somewhat idealized scenario. Simulations have been performed for clear-sky conditions, with no aerosols in the scenes. In retrievals from actual measured radiances, even for a clear-sky, non-scattering atmosphere, there is always some forward model error due to, e.g., uncertainties in spectroscopy, interfering species and the treatment of the surface. With real data, these kinds of uncertainties can lead to significant systematic errors in the retrievals, particularly for well-mixed greenhouse gases such as $CH_4$ and $CO_2$. For the simulations presented here, we have considered only the error term associated with measurement noise.

The measurement noise associated with the simulated radiance is obtained using the instrument model described in Section 3.2. The JPL GEO-IR sounder concept is configurable in terms of spectral range and spectral resolution, with a native spatial resolution that corresponds to a 2.1 km footprint on the ground. Different configurations of the instrument concept will affect the number of photons available in each channel and therefore impact the signal to noise. For a given integration time, lower spectral resolution leads to correspondingly higher SNR. The SNR of the observed radiance spectra can be increased by increasing the integration time. For geostationary observations, this leads to a trade-off between measurement noise and temporal resolution. An increase of the throughput (etendue) leads to lower noise (*Schwantes et al.*, 2002).

In retrievals from real data, higher spectral resolution can offer advantages in terms of ability to distinguish between the target molecule and interfering spectral signatures from other molecules with features in the spectral range of interest, despite the increase in measurement noise. In the results presented in this study, that advantage in reduction of systematic error is not accounted for. The SNR can also be increased by aggregating spatially. For example, aggregating four 2.1 km footprints would increase the SNR by a factor of two. Depending on the application of the measurements, there may be some advantage to trading spatial resolution for a gain in SNR.

## 5 Results

### 5.1 TATM and $H_2O$ retrievals

High spectral resolution is necessary to provide the vertically resolved TATM and $H_2O$ information critical for numerical weather prediction and for many other applications including local extreme weather conditions and global climate change. Current satellite-based TATM and $H_2O$ retrievals mainly utilize TIR spectral measurements. Here we also examine information gained from adding SWIR measurements. Tables 5 and 6 list the possible choices of frequency range for TATM and $H_2O$ retrievals. Some of these spectral ranges are used in current operational missions, while some are candidates for future missions.



We compare results for three values of spectral resolution and for two values of spatial resolution.
Examining the above DOFS tables, we see competing effects of spectral resolution (MOPD) and
measurement noise. As described in Section 3.2, the measurement noise (Noise Equivalent Spectral
Radiance, NESR) is estimated for a fixed integration time for both the 2.1 and 4.2 km ground sampling
distance (GSD) configurations. The NESR for the MOPD = 0.8 cm instrument is therefore smaller than
that for the MOPD = 2 or 5 cm instruments. Typically, however, the higher spectral resolution
instruments provide larger DOFS than the NSR instrument. For $H_2O$ retrievals, the optimal DOFS are
provided by the intermediate resolution instrument.
The differences in DOFS for the two GSD values are obvious. This shows the trade-off between
spatial resolution and retrieval vertical resolution and precision (not listed). Both GSDs provide high
precision, high vertical resolution TATM and $H_2O$ retrievals. We estimate the tropospheric vertical
resolution for TATM to be 1.5–2 km with <0.5 K precision, and for $H_2O$ to be 1–2 km with ~5%
precision. In comparison, representative tropospheric values for AIRS are 1 km for TATM and 2 km for
$H_2O$ (*Irion et al.*, 2018).
The selection of spectral regions also affects the TATM and $H_2O$ products. For example, using the
VLWIR+LWIR+MWIR domain provides much more sensitivity compared to using MWIR alone. Figure
5 shows averaging kernel plots for TATM and $H_2O$ for the 4.2 km GSD option for four spectral band
combinations: VLWIR+LWIR, MWIR, SWIR, and VLWIR+LWIR+MWIR+SWIR. The characteristics
of the TIR TATM and $H_2O$ retrievals are very similar to those obtained by currently operating
instruments. We note that the sensitivity of SWIR retrievals is mostly near the surface. Further, the
measurement noise in the SWIR was reduced by a factor of 5 in these figures by averaging 25 pixels,
thereby reducing the effective GSD to 21 km. Note that this is worse than the 15 km AIRS/CrIS native
resolution but better than the 45 km that the TATM and $H_2O$ products are typically reported on.
**5.2 Trace gas retrievals**
Among many possible detectable trace gases from the extended spectral radiance measurements, we
selected to examine profile retrieval characteristics for $O_3$, CO, $CH_4$ and $CO_2$ for the given instrument
configurations (see Table 3 for retrieval spectral ranges). Table 7 lists DOFS for the chosen trace gases
for the West Virginia scenario. Results for the FSR option are largely similar to those for the intermediate
spectral resolution instrument and are hence not shown. The DOFS in Table 7 are broadly consistent
with previously published work on species profile retrievals from satellite observations (*Beer*, 2006;
*Connor et al.*, 2008; *Deeter et al.*, 2009, 2015; *George et al.*, 2009; *Kulawik et al.*, 2010; *Worden et al.*,
2010, 2013; *Clerbaux et al.*, 2015; *Fu et al.*, 2016; *Smith and Barnet*, 2020). For a given spectral
resolution instrument, the higher DOFS in retrievals for the larger GSD case for all species are due to



the reduced measurement noise. For a given GSD, the DOFS are slightly higher for the NSR case
compared to the MOPD = 2 cm case, but the differences are small. It is worth reiterating that these
simulated retrievals represent an idealized scenario, where we assume perfect knowledge of interfering
species in the spectral range for any given target species. In this scenario, with a constant integration
time, the NSR option provides similar results to the MOPD = 2 cm option due to the trade-off between
spectral resolution and instrument noise.
Figure 6 shows averaging kernel plots for CO for MWIR- and SWIR-only scenarios and for
combined MWIR+SWIR retrievals. The combination of wavelength regions provides improved
sensitivity to the lower troposphere compared to either spectral region alone. $CO_2$ retrievals (Figure 7)
benefit the most from the combination of VLWIR+MWIR+SWIR retrievals. The SWIR domain adds
sensitivity in the lower troposphere and near the surface. The characteristics of the $CO_2$ retrievals are in
good agreement with OCO-2/3 observations. For $CH_4$ (Figure 8), the addition of SWIR bands also
provides noticeable enhancement in lower tropospheric and near-surface sensitivity. For CO retrievals,
the contribution of the SWIR to the near-surface sensitivity is less pronounced. The stronger contribution
of SWIR measurements to the total DOFS for $CH_4$ and $CO_2$ compared to CO is a result of three factors:
(1) lower top of the atmosphere solar irradiance in the CO spectral region relative to the $CH_4$ and $CO_2$
regions, (2) lower surface albedo, and (3) larger absorption, primarily by $H_2O$ and $CH_4$. Our results for
$O_3$ are broadly consistent with published results for LWIR satellite observations (e.g., *Nassar et al.*, 2008;
*Smith and Barnet*, 2020). Figures 6–8 use the same effective GSD of 21 km in the SWIR as described in
Section 5.1.

**6 Use of synthetic retrievals in Observation System Simulation Experiments**
Observing system simulation experiments (OSSEs) are used to assess the potential information in a
new set of measurements before they are deployed. In the case of satellite remote sensing, there are
several types of experiment that may be performed (*Zeng et al.*, 2020*)*: a sampling OSSE, in which an
orbit simulator is used to determine how often the observing system views features of interest (e.g.,
*Crespo et al.*, 2017); a geophysical variable (or retrieval) OSSE, in which synthetic measurements are
used to estimate synthetic geophysical variables and their uncertainties *(*e.g., *Xu et al.*, 2019 and this
manuscript*)*; and impact OSSEs, the most common of which are those that assess the effect of
assimilation of new measurements on a weather forecast (e.g., *Hoffman and Atlas*, 2016; *Posselt et al.*,
2021). This paper describes a set of geophysical variable/retrieval OSSEs, and the observing system
characteristics (spatial and temporal resolution and uncertainties).



The detailed characterization of uncertainties in the retrieved TATM and $H_2O$ retrievals provided
by this study will be directly incorporated into a set of weather forecast OSSEs, the results of which will
be reported in a subsequent paper. Note that, for a weather forecast OSSE to be credible, it is crucial to
represent the synthetic measurements as accurately as possible. TATM and $H_2O$ precision and total error
are reported in Table 8; it can be seen that the errors for the MWIR-only configuration are on the order
of the errors in CrIS and AIRS retrievals, while the full-spectrum JPL GEO-IR Sounder configuration
yields total errors that are smaller than those from either CrIS or AIRS. As such, assimilation of
information from JPL GEO-IR Sounder measurements is expected *a priori* to have as much or greater
impact on weather forecasts compared with existing hyperspectral sounders. Note that the total error in
the full-spectral-range TATM and $H_2O$ retrievals is equivalent to, or less than, the uncertainty reported
for radiosonde measurements of these quantities (*Rienecker et al.*, 2008; Table 3.5.2).
While it is common to assimilate radiances (rather than retrieved TATM and $H_2O$) in modern data
assimilation systems, this is not necessarily the right choice in an OSSE. This is because radiance
assimilation necessitates careful tuning of the radiative transfer model to remove bias and to make it
consistent with the model processes and resolution. This is an iterative and time consuming process that
requires comparison with other measurements. In addition, the true state (from the nature run) is available
in an OSSE. This means that an unbiased mapping from measurements to state space is straightforward,
in contrast to real measurements. If the TATM and $H_2O$ retrieval algorithm can be used to apply the
resolution and noise characteristics to the nature run profiles, then assimilation of retrievals is a more
straightforward and more realistic option.
Finally, we note that there will be particular advantages and challenges in assimilating the high
temporal resolution data that will be available from the JPL GEO-IR Sounder. The clear advantage is the
ability to observe rapidly evolving processes (e.g., the environment around thunderstorms and
hurricanes). This information is not available from the current LEO constellation. However, modern data
assimilation systems are configured for assimilation of intermittent data (at best hourly in operational
data assimilation systems) and will require modification to make best use of the high time frequency
geostationary soundings provided by the JPL GEO-IR Sounder.

**7 Conclusions**

In this paper, we present an end-to-end retrieval study for a proposed FTS instrument covering the
entire infrared spectral range from 1–15 μm from a geostationary satellite orbit. An instrument model is
used to derive realistic measurement radiance and noise for several diurnal observations over small
ground footprints (e.g., 2.1 km). We perform TATM and trace gas profile retrievals for the JPL GEO-IR


Sounder that covers the entire VLWIR, LWIR, MWIR and SWIR spectral domains. Retrieval
characteristics, such as DOFS and measurement error, are examined in order to evaluate the performance
of several instrument configurations. These configurations include VLWIR-, LWIR-, MWIR-, and
SWIR-only and their combinations, and different spectral and spatial resolutions, for a realistic
geostationary observing system making field-of-view observations at fixed time intervals. Two summer-
time atmospheres are used: a scenario near Houston as a clean-air case, and one in West Virginia
representing a polluted scenario. We analyze TATM, $H_2O$, $O_3$, CO, $CH_4$ and $CO_2$ profile retrievals.

High spectral resolution can provide improved ability to distinguish absorption lines of the target

species from interferents. In the case of species (such as $O_3$) where much of the total column lies in the
stratosphere, higher spectral resolution also provides enhanced ability to separate the tropospheric signal
from the stratospheric signal. When the total integration time is fixed, there is a trade-off between spectral
resolution and noise. In the idealized retrievals presented here, we assume perfect knowledge of
interfering species. In this case, three different MOPDs provide comparable results in terms of DOFS.
However, in the real world, we would expect higher spectral resolution to offer advantages in terms of
reduction in systematic errors.

Compared to single spectral region instruments, e.g., only LWIR or MWIR, combinations of

VLWIR/LWIR/MWIR/SWIR enhance the sensitivity of the retrievals to the lower troposphere. In our
analyses, we find that the contributions from the SWIR in the combined measurements are noticeable for
both trace gas and TATM retrievals, especially when the ground pixels are averaged to reduce
measurement noise in the SWIR. In particular, the SWIR measurements add information in the lower
troposphere and for near-surface species retrievals.

We limit the spatial resolution choices to GSD = 2.1 km and 4.2 km in our simulations. Especially

for multi-band retrievals, the results are realistically adequate for many research applications for both
ground sampling footprints. We compare performance metrics (e.g., NESR and SNR) for the proposed
instrument with values for several current/past satellite instruments in multiple spectral bands. The
performance of the JPL GEO-IR Sounder is similar to or better than currently operational instruments.
At the same time, the JPL GEO-IR Sounder provides much higher spatial and temporal resolution and a
wider range of trace gases than current instruments that combine TIR and SWIR. The derived retrieval
characteristics (e.g., DOFS and retrieval errors) also compare favorably with currently available
products.

**Data availability**

The code and data are available from the authors upon request.



**Author contributions**


SPS, Y-HW and LID conceived the work. VN provided the radiative transfer model, led the
simulated retrieval work, and prepared the manuscript. ML, J-FB and ZZ assisted with the retrievals. ML
provided the trace gas absorption and inverse models. JLN provided the profiles for the simulations. SSK
provided the emissivity database and advised on the retrieval constraints. J-FB provided the instrument
model. VHP and SPS helped analyze the simulation results. LW, JAR and DJP provided the connection
with OSSEs. All listed authors contributed to the review and editing of this manuscript.

**Competing interests**
The authors declare that they have no conflict of interest.

**Acknowledgements**
A portion of this research was carried out at the Jet Propulsion Laboratory, California Institute of
Technology, under a contract with the National Aeronautics and Space Administration
(80NM0018D0004). The authors acknowledge S. Kulawik for helpful discussions on retrieval
constraints.

**Financial Support**
The authors acknowledge support from the National Oceanic and Atmospheric Administration
through BAA-NOAA-GEO-2019, and the Jet Propulsion Laboratory Advanced Concepts Program.



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



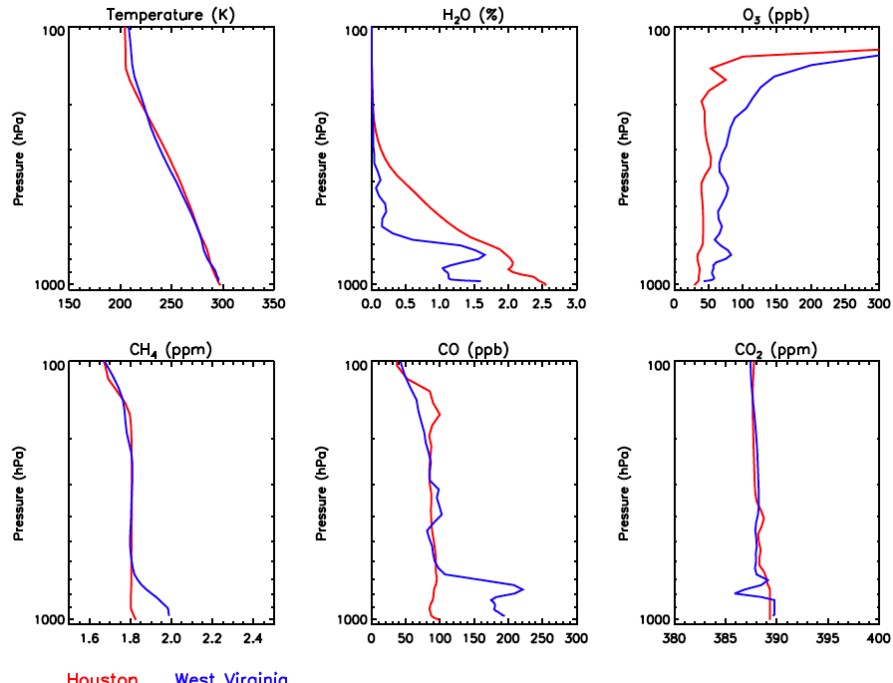


**Figure 1: Scenarios considered in the simulations.**






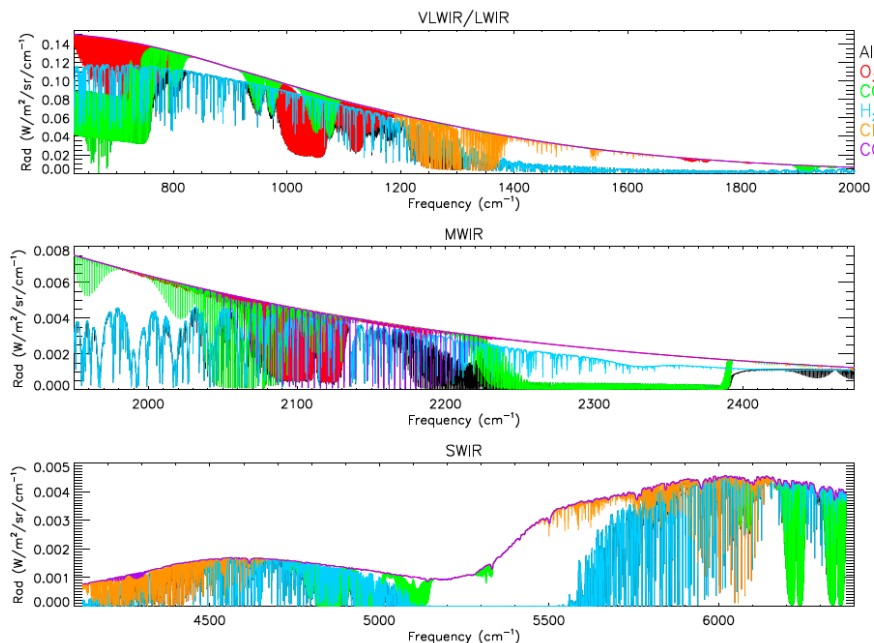


**Figure 2. Simulated top of the atmosphere monochromatic radiances (black) in the 650–7000 cm⁻¹ wavelength range for atmospheric profile near Houston. Also shown are radiances corresponding to (red) O₃, (green) CO₂, (blue) H₂O, (orange) CH₄, and (purple) CO absorption.**




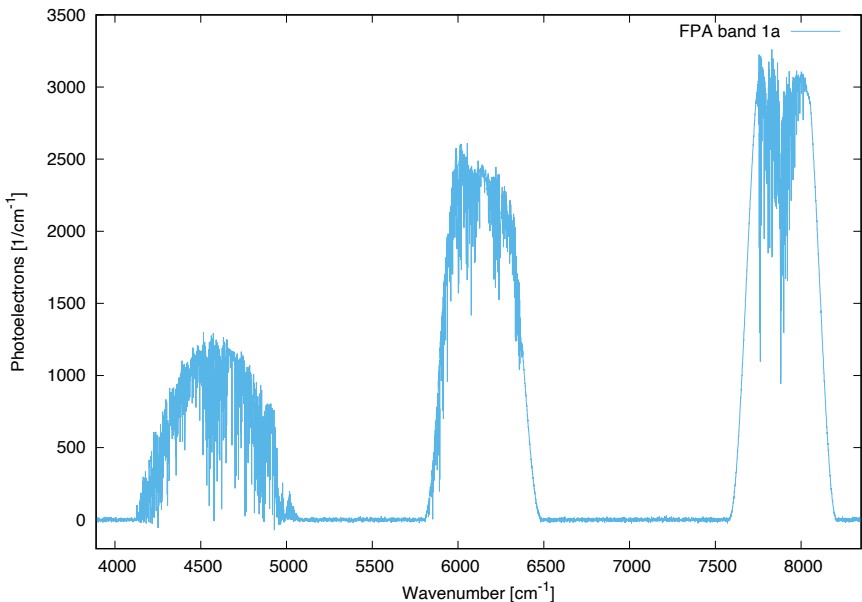


**Figure 3: Simulated JPL GEO-IR Sounder spectrum in the SWIR domain. The SWIR domain is sub-divided**

**into discrete bands using a triple-band interference filter to maximize the SNR in spectral regions of interest**

**($CO_2$, $CH_4$, CO, $H_2O$, and $O_2$).**




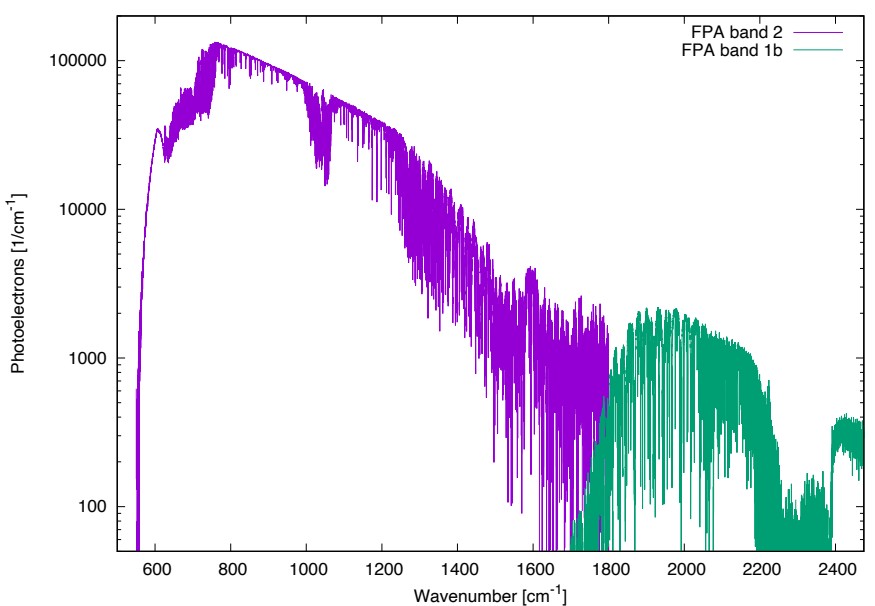


**Figure 4: Simulated JPL GEO-IR Sounder spectrum in the VLWIR, LWIR and MWIR domains. Note the**

**logarithmic scale.**


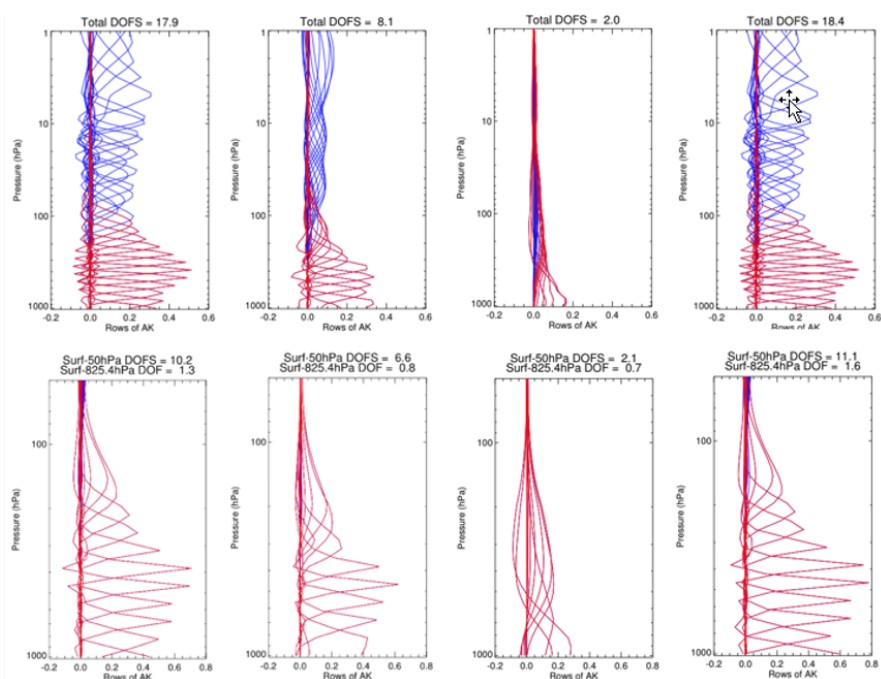


**Figure 5: Plots of averaging kernel rows for (top) TATM and (bottom) H₂O. The spectral ranges are (from**
**left to right) VLWIR+LWIR, MWIR, SWIR, and VLWIR+LWIR+MWIR+SWIR. These results are for the**
**Houston case. The blue and red lines refer to averaging kernel rows for pressure levels above and below 100**
**hPa, respectively.**






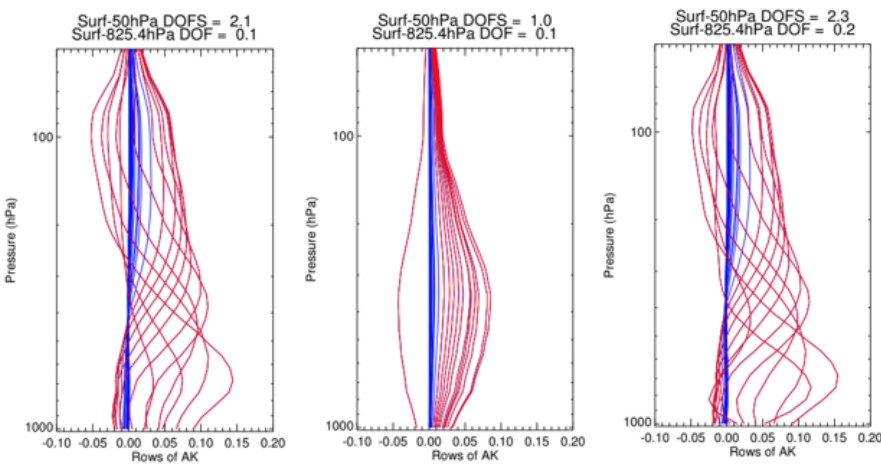

**Figure 6: Plots of averaging kernel rows for CO retrievals. The spectral ranges are (from left to right) MWIR,**
**SWIR, and MWIR+SWIR. These results are for the West Virginia case. The color scheme is the same as in**
**Figure 5.**





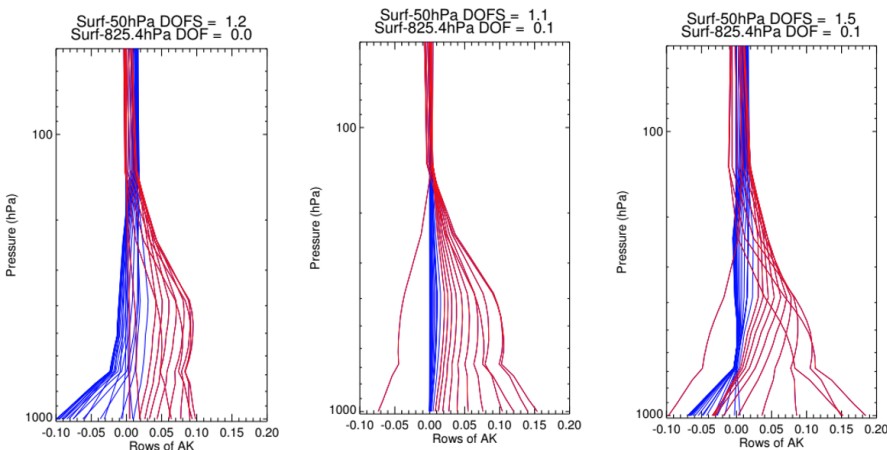

**Figure 7: Plots of averaging kernel rows for CO₂ retrievals. The spectral ranges are (from left to right)**

**VLWIR+MWIR, SWIR, and VLWIR+MWIR+SWIR. These results are for the West Virginia case. The color**

**scheme is the same as in Figure 5.**





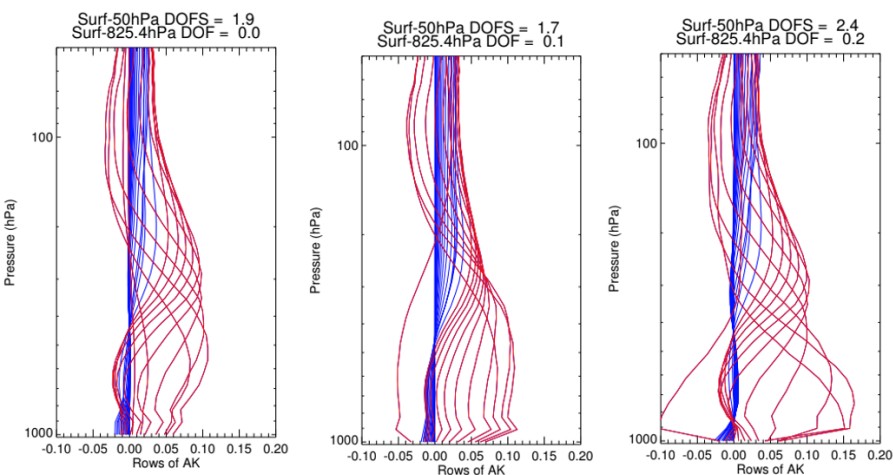

**Figure 8: Plots of averaging kernel rows for CH₄ retrievals. The spectral ranges are (from left to right) LWIR, SWIR, and LWIR+SWIR. These results are for the West Virginia case. The color scheme is the same as in Figure 5.**



**Table 1. Spectral ranges and their designations used in this study.**

| Designation | Spectral Range (µm) | Spectral Range (cm$^{-1}$) |
|---|---|---|
| VLWIR | >10 | <1,000 |
| LWIR | 5–10 | 1,000–2,000 |
| MWIR | 3–5 | 2,000–3,333 |
| SWIR | 1–3 | 3,333–10,000 |
| TIR | >3 | <3,333 |






**Table 2. Current and planned missions making spaceborne, spectrally resolved measurements of TIR and**
**SWIR radiances. Note that MOPITT was designed to also offer measurements of CH$_4$, although that did not**
**materialize (hence the gray shading).**

| Orbit | Instrument/mission | T profile | H$_2$O | | CO | | CH$_4$ | | CO$_2$ | |
|---|---|---|---|---|---|---|---|---|---|---|
| | | TIR | TIR | SWIR | TIR | SWIR | TIR | SWIR | TIR | SWIR |
| LEO | Hyperspectral TIR sounders (AIRS, CrIS, IASI, IASI-NG, TES) | Y | Y | | Y | | Y | | Y | |
| | MOPITT | | | | Y | Y | | | | |
| | GOSAT, GOSAT-2 | Y | Y | Y | Y | | Y | Y | Y | Y |
| | OCO-2/OCO-3 | | | Y | | | | | | Y |
| | TROPOMI | | | Y | | Y | | Y | | |
| | TANSAT | | | Y | | | | | | Y |
| GEO | IRS | Y | Y | | Y | | Y | | Y | |
| | GIIRS | Y | Y | | Y | | Y | | Y | |
| | GeoCarb | | | Y | | Y | | Y | | Y |
| | JPL GEO-IR Sounder | Y | Y | Y | Y | Y | Y | Y | Y | Y |





**Table 3. Spectral ranges used in this study for simulated retrievals of CO, CH₄ and CO₂.**

| Molecule | Spectral Ranges (cm⁻¹) | Relevant For |
|---|---|---|
| Carbon monoxide (CO) | 2000–2250<br>4210–4350 | Air quality and carbon cycle (combustion and fire emissions) |
| Methane (CH₄) | 1210–1380<br>4210–4350<br>6000–6150 | Greenhouse gas monitoring and carbon cycle (wetlands, oil and gas, agriculture) |
| Carbon dioxide (CO₂) | 650–1100<br>2250–2450<br>4810–4900<br>6170–6290 | Greenhouse gas monitoring and carbon cycle (human emissions, status of land and ocean carbon sinks) |




**Table 4. Comparison of JPL GEO-IR Sounder with other state-of-the-art instruments.**

| Instrument | GIIRS | IRS | CrIS | JPL GEO-IR Sounder |
|---|---|---|---|---|
| Status | In space | 2023 launch | In space | This study |
| Nationality | China | EU | US | US |
| Orbit | GEO | GEO | Polar | GEO |
| Longitude (°) | 104.7 E | 0–45 E | N/A | 75–137 W |
| Spacecraft | Dedicated | Dedicated | Dedicated | Hosted payload |
| GSD, nadir (km) | 16 | 4 | 14 | 4.2 (binned), 2.1 (native) |
| Spectral range (cm$^{-1}$ unless otherwise indicated) | 700–1130 1650–2250 0.55–0.75 μm | 680–1210 1600–2250 | 650 – 1095 1210 – 1750 2155 – 2550 | 650–10,000* |
| Resolution (cm$^{-1}$) | 0.625 | 0.625 | 0.625 | NSR** = 0.625, FSR = 0.1 |
| Full Disk Revisit Time (hr) | 2–3 | 1 | 12 | 0.2 |

*FTS instrument capability
** NSR = Nominal Spectral Resolution. FSR = Full Spectral Resolution. FSR mode decreases retrieval biases
caused by interfering absorbers



**Table 5. DOFS for TATM retrievals for three spectral (MOPD) and two spatial (GSD) resolution scenarios.**
**The values shown here are for the Houston profile.**

| Frequency Domain | DOFS (MOPD = 5 cm) | | DOFS (MOPD = 2 cm) | | DOFS (MOPD = 0.8 cm) | |
|---|---|---|---|---|---|---|
| | 2.1 km GSD | 4.2 km GSD | 2.1 km GSD | 4.2 km GSD | 2.1 km GSD | 4.2 km GSD |
| VLWIR+LWIR | 13.6 | 17.6 | 14.2 | 17.9 | 14.3 | 17.9 |
| MWIR | 5.1 | 7.9 | 5.8 | 8.3 | 6 | 8.1 |
| VLWIR+LWIR+ MWIR | 13.8 | 17.8 | 14.4 | 18.1 | 14.5 | 18.1 |
| SWIR | 0.2 | 1.6* | 0.3 | 1.8* | 0.4 | 2.0* |
| VLWIR+LWIR+ MWIR+SWIR | 13.8 | 17.9* | 14.6 | 18.3* | 14.7 | 18.4* |

* Instrument noise is reduced by a factor of 5 through footprint averaging for the SWIR only,
providing an effective GSD of 21 km.



**Table 6: Same as Table 5 but for H₂O**

| Frequency Domain | DOFS (MOPD = 5 cm) | | DOFS (MOPD = 2 cm) | | DOFS (MOPD = 0.8 cm) | |
|---|---|---|---|---|---|---|
| | 2.1 km GSD | 4.2 km GSD | 2.1 km GSD | 4.2 km GSD | 2.1 km GSD | 4.2 km GSD |
| VLWIR+LWIR+ | 7.9 | 11.2 | 8.2 | 11.3 | 8.2 | 11.2 |
| MWIR | 4.6 | 6.9 | 5.0 | 7.3 | 4.6 | 6.6 |
| VLWIR+LWIR+ MWIR | 8.3 | 11.8 | 8.8 | 12.1 | 8.6 | 11.9 |
| SWIR | 1.2 | 2.2* | 1.3 | 2.1* | 1.4 | 2.1* |
| VLWIR+LWIR+ MWIR+SWIR | 8.3 | 12.1* | 8.9 | 12.3* | 8.7 | 12.1* |

\* Instrument noise is reduced by a factor of 5 through footprint averaging for the SWIR only,

providing an effective GSD of 21 km.



**Table 7. Trace gas retrieval configurations and DOFS for the West Virginia profile. TATM and H₂O are**
**simultaneously retrieved when listed.**

| Retrieved Species | Frequency Domain | DOFS (MOPD = 2 cm) | | DOFS (MOPD = 0.8 cm) | |
|---|---|---|---|---|---|
| | | 2.1 km GSD | 4.2 km GSD | 2.1 km GSD | 4.2 km GSD |
| $O_3$ (TATM, $H_2O$) | LWIR | 3.5 | 4.0 | 3.4 | 4.0 |
| CO | MWIR | 1.7 | 2.1 | 1.6 | 2.1 |
| | SWIR | 0.08 | 0.96* | 0.1 | 0.96* |
| | MWIR+SWIR | 1.7 | 2.3* | 1.7 | 2.3* |
| $CH_4$ (TATM, $H_2O$) | LWIR | 1.5 | 2.0 | 1.6 | 2.1 |
| | SWIR | 0.7 | 1.9* | 0.8 | 1.9* |
| | LWIR+SWIR | 1.6 | 2.7* | 1.8 | 2.8* |
| $CO_2$ (TATM, $H_2O$) | VLWIR | 1.0 | 1.5 | 1.1 | 1.6 |
| | VLWIR+MWIR | 1.0 | 1.5 | 1.2 | 1.6 |
| | SWIR | 0.3 | 1.1* | 0.4 | 1.1* |
| | VLWIR+MWIR +SWIR | 1.0 | 1.7* | 1.2 | 1.9* |

* Instrument noise is reduced by a factor of 5 through footprint averaging for the SWIR only,
providing an effective GSD of 21 km.



**Table 8. Estimates of total and precision errors for JPL GEO-IR Sounder, CrIS and AIRS TATM and H$_2$O**
**retrievals in the troposphere. Note that data used for CrIS and AIRS retrievals were obtained near Houston,**
**Texas in August 2020. Averaged retrieved cloud optical depths are limited to less than 0.1, consistent with**
**mostly clear-sky conditions.**

|  | TATM | | H$_2$O (lower-mid troposphere) | |
|---|---|---|---|---|
|  | Total Error | Precision | Total Error | Precision |
| JPL GEO-IR Sounder (MWIR-only) | 0.5–1.5 K | 0.2–0.6 K | ~8% | ~5% |
| JPL GEO-IR Sounder (Entire spectral range) | 0.3–1 K | 0.1–0.3 K | ~5% | ~3% |
| CrIS | 0.5–1.5 K | 0.2–0.3 K | 10–13% | 2–3% |
| AIRS | 0.5–1.2 K | ~0.3 K | 15–30% | 2–5% |
