# Peer review of "Simulated Multispectral Temperature and Atmospheric Composition Retrievals for the JPL GEO-IR Sounder"

_Atmospheric Measurement Techniques, 2021_

## Author Comment (AC1)

We would like to thank the reviewer for their thoughtful and constructive comments and suggestions to improve the clarity of the manuscript. We have made changes to address these comments and suggestions.

Point-by-point responses to the comments are provided below. The reviewer comments are in blue, our responses are in red (line numbers refer to those in the revised manuscript), and modifications to the original manuscript are highlighted in yellow.

Sincerely,

Vijay Natraj

On behalf of all co-authors

**RC1: 'Comment on amt-2021-290', Anonymous Referee #1, 18 Nov 2021**

The authors present a detailed computation of DOFs for atmospheric retrievals for a notional multispectral sounder. The paper is well presented and high-quality a-priori data that has been used for other instruments is used for their DOF estimates. I don't think their are any surprises here, but the paper does give decisions makers and proposers a feel for the benefits of their notional sounder (most likely a very expensive one indeed).

However, I think the discussion on OSSEs near the end is uneeded and gives the reader the wrong impression. No OSSEs are done here and I don't think it is warranted to talk about what you are going to do in the future. A sentence will do.

We thank the reviewer for this useful feedback. We agree that the section on OSSEs looks redundant because such experiments were not performed in this work. However, we do feel that the utility of the retrievals for OSSEs is an important issue that deserves discussion in the manuscript. Therefore, we have renamed Section 6 and shortened the discussion as follows (lines 360–386):

**6 Discussion: Use of GEO-IR Information in Data Assimilation and Observation System Simulation Experiments**

We have focused in this paper on the characteristics of the measurements and retrievals that we expect to obtain from the GEO-IR observing platform. While this paper does not deal directly with the use of this information in a data assimilation system, the results we have presented lay the necessary groundwork for future work in this area. In particular, the detailed characterization of uncertainties in the TATM and $H_2O$ retrievals provided by this study can be directly incorporated into a set of weather forecast OSSEs. We have begun this research, and will report on the results in a subsequent paper. Note that, for a weather forecast OSSE to be credible, it is crucial to represent the synthetic measurements as accurately as possible. TATM and $H_2O$ precision and total error are reported in Table 8; it can be seen that the errors for the MWIR-only configuration are on the order of the errors in CrIS and AIRS retrievals, while the full-spectrum JPL GEO-IR Sounder configuration yields total errors that are smaller than those from either CrIS or AIRS. As

such, assimilation of information from JPL GEO-IR Sounder measurements is expected a priori to have as much or greater impact on weather forecasts compared with existing hyperspectral sounders. Note that the total error in the full-spectral-range TATM and $H_2O$ retrievals is equivalent to, or less than, the uncertainty reported for radiosonde measurements of these quantities (Rienecker et al., 2008; Table 3.5.2).

We also note that there will be particular advantages and challenges in assimilating the high temporal resolution data that will be available from the JPL GEO-IR Sounder. The clear advantage is the ability to observe rapidly evolving processes (e.g., the environment around thunderstorms and hurricanes). This information is not available from the current LEO constellation. However, many modern data assimilation systems are configured for assimilation of intermittent data (at best hourly in operational data assimilation systems). While four-dimensional variational data assimilation (4D-Var) is capable of ingesting data at non-synoptic times, assimilation of sub-hourly data remains challenging. It is likely that all but the most rapid-update data assimilation systems will require modification to make best use of the high time frequency geostationary soundings provided by the JPL GEO-IR Sounder.

---

## Author Comment (AC2)

We would like to thank the reviewer for their thoughtful and constructive comments and suggestions to improve the clarity of the manuscript. We have made changes to address these comments and suggestions.

Point-by-point responses to the comments are provided below. The reviewer comments are in blue, our responses are in red (line numbers refer to those in the revised manuscript), and modifications to the original manuscript are highlighted in yellow.

Sincerely,

Vijay Natraj

On behalf of all co-authors

**RC3: 'Comment on amt-2021-290', Anonymous Referee #3, 30 Dec 2021**

General comments:

Interesting and informative article, especially that it includes temperature, moisture and trace gases. There are many benefits to a full spectral (VIS/NIR/IR/LWIR) coverage instrument.

While it's stated in the manuscript, there is no mention of deriving wind fields in the abstract. Please add something about winds in the abstract, as it's a key to what the geostationary perspective supplies.

We modified the first sentence of the abstract to address the reviewer's comment (lines 17–18):

Satellite measurements enable quantification of atmospheric temperature, humidity, wind fields, and trace gas vertical profiles.

Please consider including these references:

https://journals.ametsoc.org/view/journals/atot/26/11/2009jtecha1248_1.xml (JTECH Why geo sounder?)

We have added the reference (lines 66–68):

Measurements from geostationary (GEO) orbit can provide contiguous horizontal (~4 km) and temporal (full sounding disk coverage in 1–2 hours) resolution not possible from LEO (e.g., Schmit et al., 2009).

https://agupubs.onlinelibrary.wiley.com/doi/10.1029/2021GL093672 (FY-4A GIIRS winds)

We have added the reference (lines 87–88):

- Capability for retrievals of 4-D winds from combinations of TATM and $H_2O$ temporal imagery as recently described using GIIRS data (Ma et al., 2021; Yin et al., 2021)

http://www.iapjournals.ac.cn/aas/en/article/doi/10.1007/s00376-018-8036-3 (AAS Local storm OSSE)

We have added the reference (lines 379–381):

The clear advantage is the ability to observe rapidly evolving processes (e.g., the environment around thunderstorms and hurricanes; see, e.g., Li et al., 2018).

https://journals.ametsoc.org/view/journals/bams/102/5/BAMS-D-19-0304.1.xml (BAMS MTG)

https://repository.library.noaa.gov/view/noaa/32921 (NOAA Value Assessment)

https://journals.ametsoc.org/view/journals/bams/98/8/bams-d-16-0065.1.xml (BAMS FY-4A)

Given that an advanced IR sounder is slated for the GeoXO mission (https://www.nesdis.noaa.gov/next-generation-satellites/geostationary-extended-observations-geoxo) should be noted.

We thank the reviewer for these references, and have added them (lines 68–74):

The IRS instrument onboard the Meteosat Third Generation Sounder platform will track the four-dimensional structure of TATM and $H_2O$ (Holmlund et al., 2021). The GIIRS instrument on the Fengyun-4 meteorological satellite has similar capabilities (Yang et al., 2017). Adkins et al. (2021) describe in comprehensive detail the value of a hyperspectral IR sounder in GEO orbit. Based on this report, an advanced high-resolution IR sounder has been recommended for the Geostationary Extended Observations (GeoXO) mission (https://www.nesdis.noaa.gov/next-generation-satellites/geostationary-extended-observations-geoxo).

Detailed comments:

Line 63. At least add Suomi to Suomi-NPP.

Done.

Line 64. Assume you are taking about geo sounders, if so, that should be stated. Don't think that 1 hour refresh is a fine as what geo can provide (ABI provides 1min data). FY-4A GIIRS has provided 15 min data over regional areas. If you are referring to only full disk (or full sounding disk) coverage, than that should be stated.

Yes, we are talking about GEO sounders. We are also referring to full sounding disk coverage, as the reviewer inferred. Our baseline is projection of a 512×512 array at ~4 km/pixel at nadir. Each projection will require several minutes, so the full disk will require ~1–2 hours. We have modified the sentence for increased clarity (lines 66–68):

Measurements from geostationary (GEO) orbit can provide contiguous horizontal (~4 km) and temporal (full sounding disk coverage in 1–2 hours) resolution not possible from LEO (e.g., Schmit et al., 2009).

Line 67. Consider to add a winds column for Table 2. Would be "yes" for the geo's.

Done.

Figures 2-4. Please add wavelength labels along the top of each plot. Many 'think' in this space and not cm-1.

Done.

Line 113. How does this database compare to the IREMIS database (https://cimss.ssec.wisc.edu/iremis/)? (in overlapping spectral regions).

We have not done a comparison with the IREMIS database. That is outside the scope of this study. However, as noted, IREMIS only seems to cover wavelengths longer than 3.8 microns. Our database provides continuous emissivity values from 250 nm to 20 microns.

Line 263 seems wordy. Maybe delete that sentence and modify the following sentence:

Simulations have been performed for the idealized clear-sky/no aerosols conditions.

Done.

Line 269. Remove 'here'.

We feel that it sounds awkward without "here", so we decided not to make the suggested change.

Line 315. If you average 25 pixels, might you bring in more (un-detected) cloud "noise"?

The plan is to use a 2K×2K array, binned to 512×512. With this set-up, we anticipate that some cloud screening could be done on-board based on the radiances. However, that is getting too far into the weeds for this manuscript. The advantage of our approach is that, while 5×5 pixels may be required for the CO retrievals and is therefore a little worse than AIRS/CrIS, we measure TIR and SWIR at the same time, eliminating bias from observing with separate instruments. Further, as noted in the paper: "Note that this is worse than the 15 km AIRS/CrIS native resolution but better than the 45 km that the TATM and $H_2O$ products are typically reported on." We have added a sentence addressing this point (lines 326–329):

Further, while 5×5 pixels may be required for trace gas retrievals in the SWIR (see section 5.2) and is therefore a little worse than AIRS/CrIS, we measure TIR and SWIR at the same time, eliminating bias from observing with separate instruments.

Line 384. Isn't a 4d-var approach supposed to better handle more frequent observations?

The reviewer is correct that 4D-Var systems are capable of processing rapid-in-time observations, and we have modified the text to reflect this (lines 381–387).

However, many modern data assimilation systems are configured for assimilation of intermittent data (at best hourly in operational data assimilation systems). While four-dimensional variational data assimilation (4D-Var) is capable of ingesting data at non-synoptic times, assimilation of sub-hourly data remains challenging. It is likely that all but the most rapid-update data assimilation systems will require modification to make best use of the high time frequency geostationary soundings provided by the JPL GEO-IR Sounder.

Table 4. Should include the 2nd GIIRS as well, or at least be clear that table 4 refers to the demo unit.

It seems that the first GIIRS that the reviewer is referring to is the one on the FY-4A satellite that had 16km GSD, and the second one is that on the FY-4B satellite, which has 12 km GSD (as do the follow-ons). We changed the GSD entry for GIIRS in Table 4 to "16 km (prototype), 12 km (follow-ons)".